# FAIR-Ensemble: Homogeneous Deep Ensembling Naturally Attenuates Disparate Group Performances

## Abstract

Ensembling multiple Deep Neural Networks (DNNs) is a simple and effective way to improve top-line metrics and to outperform a larger single model. In this work, we go beyond top-line metrics and instead explore the impact of ensembling on subgroup performances. Surprisingly, we observe that even with a simple homogeneous ensemble –all the individual DNNs share the same training set, architecture, and design choices– the minority group performance disproportionately improves with the number of models compared to the majority group, i.e. fairness naturally emerges from ensembling. Even more surprising, we find that this gain keeps occurring even when a large number of models is considered, e.g. 20, despite the fact that the average performance of the ensemble plateaus with fewer models. Our work establishes that simple DNN ensembles can be a powerful tool for alleviating disparate impact from DNN classifiers, thus curbing algorithmic harm. We also explore why this is the case. We find that even in homogeneous ensembles, varying the sources of stochasticity through parameter initialization, mini-batch sampling, and data-augmentation realizations, results in different fairness outcomes.

## 1 Introduction

Deep Neural Networks (DNNs) are powerful function approximators that outperform other alternatives on a variety of tasks (Vaswani et al., 2017; Arulkumaran et al., 2017; Hinton et al., 2012; He et al., 2016b). To further boost performance, a simple and popular recipe is to average the predictions of multiple DNNs, each trained independently from the others to solve the given task, this is known as *model ensembling* (Breiman, 2001; Dietterich, 2000).

By averaging independently trained models, one avoids single model symptomatic mistakes by relying on the wisdom of the crowd to improve generalization performance, regardless of the type of model being employed. While existing work has focused on improvements towards aggregate performance (Fort et al., 2019; Gupta et al., 2022; Opitz & Maclin, 1999) or gains in efficiency over a single larger model (Wang et al., 2020; Wortsman et al., 2022), there has been limited consideration of how sensitive ensembling performance is on certain subsets of the data distribution.

Understanding performance on subgroups is a frequent concern from a fairness perspective. A common fairness objective is mitigating disparate impact (Kleinberg et al., 2016; Zafar et al., 2015) where a class or subgroup of the dataset presents far higher error rates than other subsets of the distribution. In particular, and as we will thoroughly describe in Sec. 2, many strategies have emerged to improve fairness by designing novel ensembling strategies based on fairness measures obtained from labeled attributes. In this study, we take a step back and focus on studying the fairness benefits of the simplest ensembling strategy: homogeneous ensembles. In this setting, the individual models in the ensemble all have the same architecture and hyperparameters. They are also trained with the same optimizer, data-augmentations, and training set.

Our results are surprising: despite the absence of "diversity" in the models being trained in the homogeneous ensemble, the only sources of randomness are (i) the parameters' initialization, (ii) the realizations of the data-augmentations, and (iii) the ordering of the mini-batches. The final predictions are diverse enough to provide substantial improvements for both the minority groups and the

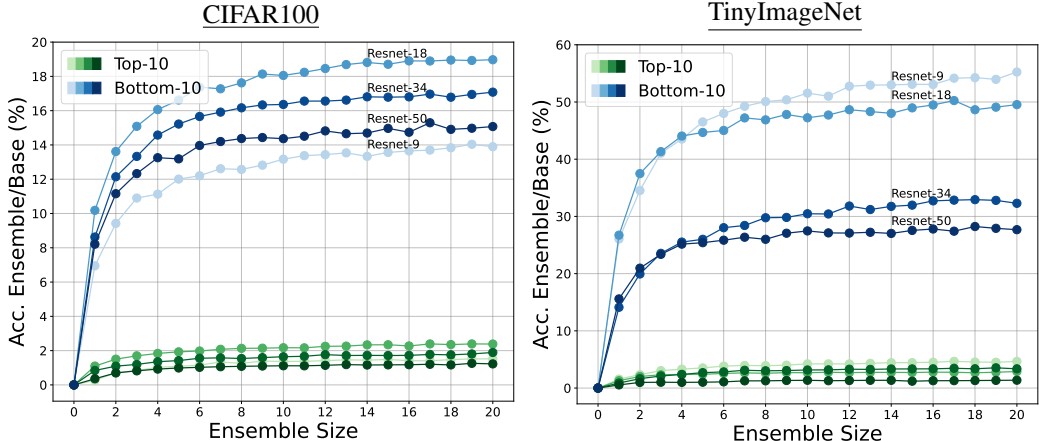

Figure 1: Relative Accuracy for Top-K/Bottom-K. Plot of the ratio of the homogeneous ensemble accuracy over a single base model (**y-axis**) illustrates strong benefits for the bottom-k group of ensembling while the top-k group only marginally benefits.

bottom-k classes upon which a single model performs badly. This emergence of fairness is observed consistently across thousands of experiments on popular architectures (ResNet9/18/34/50, VGG16, MLPMixer, ViTs) and datasets (CIFAR10/100/100-C, TinyImagenet, CelebA) (Sec. 3). The first important conclusion unlocked by our thorough empirical validation is that one may effectively improve minority group performance by using the same architecture and hyperparameters for each individual model without the need to observe corresponding labeled attributes. A second crucial finding is that solely controlling for initialization, batch ordering, and data-augmentation realizations is already enough to make training episodes produce models that are complementary with each other. Other factors such as architectures, optimizers, or data-augmentation families may not be the most important variables to produce fair ensemble (Sec. 4). The last interesting observation is that, as a function of the number of models in the homogeneous ensemble, the average performance quickly plateaus after 4 to 5 models, but the bottom-k group performance keeps increasing steadily for up to 50 models. In short, when performing deep ensembling, one should employ as many models as possible–even beyond the point at which the average performance plateaus–in order to produce a final ensemble with as much fairness as possible. Beyond fairness of homogeneous deep ensembles, our empirical study also offers a rich variety of new observations e.g., tying the severity of image corruption to the relative benefits that emerges from homogeneous deep ensembles.

**Our contributions can be enumerated as follows**:

1. We demonstrate that simple homogeneous deep ensembles trained with the same objective, architecture and optimization settings minimize worst-case error. This holds in both balanced and imbalanced datasets with protected attributes that the model is not trained on.

2. We further perform controlled sensitivity experiments where constructed class imbalance and data perturbation is applied (Sec. 3). We observe that homogeneous ensembles continue to improve fairness and, in particular, the bottom-k group benefits more and more with the size of the ensemble compared to the top-k group as the severity of the corruption increases. These observations are held even when the protected attribute is imbalanced and underrepresented, such as in our CelebA experiments.

3. We further dive into possible causes for this emergence of fairness in homogeneous deep ensembles by measuring model disagreement (Sec. 4.1) and by ablating for the different sources of randomness, e.g., weight-initialization (Sec. 4.2). We obtain interesting results that suggest certain sources of stochasticity such as mini-batch ordering or data-augmentation realizations are enough to bring diversity into homogeneous ensembles.

The codebase to reproduce our results and figures will be released upon completion of the review process.

## 2   RELATED WORK

Deep ensembling of Deep Neural Networks (DNNs) is a popular method to improve top-line metrics (Lakshminarayanan et al., 2016). Several works have sought to further improve aggregate performance by amplifying differences between models in the ensemble ranging from varying the data augmentation used for each model (Stickland & Murray, 2020), the architecture (Zaidi et al., 2021), the hyperparameters (Wortsman et al., 2022), and even the training objectives (Jain et al., 2020). *As will become clear, our focus is on the opposite setting where all the models in the ensemble share the same objective, training set, architecture, and optimizer*.

**Beyond Top-line metrics** Discussions of algorithmic bias often focus on datasets collection and curation (Barocas et al., 2019; Zhao et al., 2017; Shankar et al., 2017), with limited work to-date understanding the role of model design or optimization choices on amplifying or curbing bias (Ogueji et al., 2022; Hooker et al., 2019; Balestriero et al., 2022). Consistent with this, there has been limited work to-date on understanding the implications of ensembling on subgroup error. (Grgić-Hlača et al., 2017) points out the theoretical possibility of using an ensemble of randomly selected candidate models to improve fairness, however no empirical validation was presented. (Bhaskaruni et al., 2019) considers AdaBoost (Freund & Schapire, 1995) ensembles and shows that upweighting unfairly predicted examples reaches higher fairness. (Kenfack et al., 2021; Chen et al., 2022) propose explicit schemes to induce fairness by designing heterogeneous ensembles, and (Gohar et al., 2023) provides ensemble design suggestions in heterogeneous ensembles. Lastly, (Cooper et al., 2023) provided a modified bagging solution, again specifically designed to reduce subgroup error rate disparities. *In contrast, our goal is to demonstrate how the simplest homogeneous ensembling strategy where each model is trained independently and with identical settings naturally exhibit fairness benefits without having to measure or have labels for the minority attributes*.

**Understanding why ensembling benefits subgroup performance.** Several works to date have sought to understand why weight averaging performs well and improves top-line metrics (Gupta et al., 2022). However, few to our knowledge have sought to understand why ensembles disproportionately benefit bottom-k and minority group performance. In particular, (Rame et al., 2022) explores why weight averaging performs well on out-of-distribution data, relating variance to diversity shift. *In this work, we instead explore how individual sources of inherent stochasticity in uniform homogeneous ensembles impact subgroup performance*.

In this work, we consider the impact of ensembling on both *balanced* and *imbalanced* subgroups. Fairness considerations emerge for both groups. Real world data tends to be imbalanced, where infrequent events and minority groups are under-represented in the data collection processes. This leads to representational disparity (Hashimoto et al., 2018) where the under-represented group consequently experiences higher error rates. Even when training sets are balanced, with an equivalent number of training data points, certain features may be imbalanced leading to a long-tail within a balanced class. Both settings can result in *disparate impact*, where error rates for either a class or a subgroup are far higher (Chatterjee, 2020; Feldman & Zhang, 2020). This notion of unfairness is widely documented in machine learning systems: (Buolamwini & Gebru, 2018) find that facial analysis datasets reflect a preponderance of lighter-skinned subjects, with far higher model error rates for dark skinned women. (Shankar et al., 2017) show that models trained on datasets with limited geo-diversity show sharp degradation on data drawn from other locales. Word frequency co-occurrences within text datasets frequently reflect social biases relating to gender, race and disability (Garg et al., 2017; Zhao et al., 2018; Bolukbasi et al., 2016; Basta et al., 2019).

In the following Sec. 3, we will study how the randomness stemming from the random initialization, data-augmentation realization, or mini-batch ordering during training may provide enough diversity in homogeneous deep ensembles for fairness to naturally emerge. The why is left for Sec. 4.

## 3   FAIR-ENSEMBLE: WHEN HOMOGENEOUS ENSEMBLES DISPROPORTIONATELY BENEFIT MINORITY GROUPS

Throughout our study, we will consider a DNN to be a mapping $f_\theta : \mathcal{X} \mapsto \mathcal{Y}$ with trainable weights $\theta \in \Theta$. The training dataset $\mathcal{D}$ consists of $N$ data points $\mathcal{D} = \{\mathbf{x}_n, y_n\}_{n=1}^N$. Given the training dataset $\mathcal{D}$, the trainable weights are optimized by minimizing an objective function. We

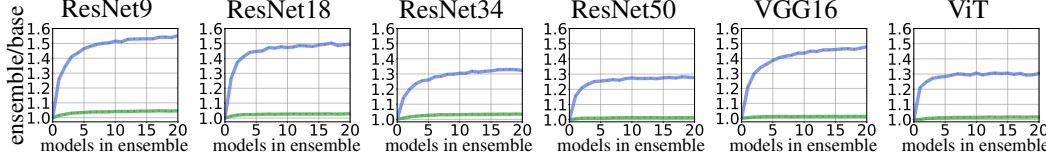

Figure 2: Test set accuracy gain as a ratio of ensemble accuracy % over the singular base model (**y-axis**) by group (top-k and bottom-k) for TinyImageNet for different architectures (**columns**) with varying the number of models within the homogeneous ensemble grows (**x-axis**). We clearly observe that as the number of models within the homogeneous ensemble grows, the bottom-k group performance improves. In particular, the bottom-k group's accuracy gain outgrow the top-k group's. This occurs despite the fact that the models within the ensemble are all employing the same hyperparameters, thus inherently share the same functional biases. The absolute accuracies are provided in Tab. 1 below, and CIFAR100 results are in Fig. 8. For the test set accuracy performance between the top-k and the bottom-k groups over ensemble size, please refer to Fig. 10 and Fig. 11.

denote a homogeneous ensemble of $m$ classification models by $\{f_{\theta_1}, \ldots, f_{\theta_m}\}$, where $f_{\theta_i}$ is the $i^{th}$ model. Each model is trained independently of the others. *We will denote by **homogeneous ensemble** the setting where the same model architecture, hyperparameters, optimizer, and training set are employed for each model of the ensemble.*

### 3.1 EXPERIMENTAL SET-UP

**Experimental set-up:** we evaluate homogeneous ensembles on CIFAR100 (Krizhevsky et al., 2009) and TinyImageNet (Russakovsky et al., 2015) datasets across various architectures: ResNet9/18/34/50 (He et al., 2016a), VGG16 (Simonyan & Zisserman, 2014), MLP-Mixer (Tolstikhin et al., 2021) and ViT (Dosovitskiy et al., 2020). Training and implementation details are provided in Appendix B. Whenever we report results on the homogeneous ensemble, unless the number of models is explicitly stated, it will comprise of 20 models. Each model is trained independently as in(Breiman, 2001; Lee et al., 2015), i.e. we do not control for any of the remaining sources of randomness as this will be explored exclusively within Sec. 4.2.

**Balanced Dataset Sub-Groups:** for top-k and bottom-k, we calculate the class accuracy of the base model and find the best and worst $K$ ($K = 10$) performing classes and track the associated classes as bottom-k and top-k groups. We then proceed to measure how performance on these groups changes as a function of the homogeneous ensemble size. We highlight that although we leverage $K = 10$ in many experiments, the precise choice of $K$ does not impact our findings, as demonstrated in Fig. 9 and Fig. 12.

**Imbalanced Dataset Sub-Groups:** we consider a setting where the protected attribute is an underlying variable different from the classification target. Similar to the setup in (Hooker et al., 2019; Veldanda et al., 2022), we treat CelebA(Liu et al., 2015) as a binary classification problem where the task is predicting hair color $\mathcal{Y} =$ {blonde, dark haired} and the sensitive attribute is gender. In this dataset, blonde individuals constitute only 15% of which a mere 6% are males. Hence, blonde male is an underrepresented attribute. We then proceed to measure how performance on the protected **gender:male** attribute varies as a function of ensemble size.

Given the above experimental details, we can now proceed to present our core observations that tie the homogeneous ensemble size with its fairness benefits.

### 3.2 OBSERVING DISPROPORTIONATE BENEFITS FOR BOTTOM-K GROUPS

**Impact on bottom-k classes:** in Fig. 1 and Fig. 2, we plot the relative gain in accuracy, i.e., the ratio between the homogeneous ensemble and base model performance on top-k/bottom-k groups, for each model architecture and dataset. Therefore answering the question: *what is the relative improvement in performance of using a homogeneous ensemble over a single model?* Across models and datasets, there is a disproportionate benefit for the bottom-k performance. For CIFAR100, this benefit ranges from 14%-29% for bottom-k across different architectures compared to 1%-4% for top-k. For TinyImageNet the benefits are even more pronounced with a maximum gain of 55% for bottom-k compared to 5% for top-k across different architectures. We also provide in Tab. 1 the absolute per-group accuracy and average performances for the corresponding models and datasets. For example, we observe a gain of more than 10% in absolute accuracy for the bottom-

Table 1: Depiction of the average and per-group (top-k and bottom-k) absolute test set accuracies corresponding to the models and datasets depicted in Fig. 2 above and Fig. 8 in the Appendix, again the homogeneous ensemble consists of 20 models. We clearly observe that fairness naturally emerges through ensembling i.e. the bottom-k group substantially benefits from homogeneous ensembling compared to the top-k group.

| | CIFAR100 | | | | | | TinyImageNet | | | | | |
| | Ensemble | | | Single | | | Ensemble | | | Single | | |
| Arch. | mean | top-k | bottom-k | mean | top-k | bottom-k | mean | top-k | bottom-k | mean | top-k | bottom-k |
|---|---|---|---|---|---|---|---|---|---|---|---|---|
| ResNet9 | 77.01 | 92.18 | 58.43 | 72.21 | 90.80 | 51.30 | 58.29 | 86.66 | 23.60 | 50.71 | 82.80 | 15.20 |
| ResNet18 | 78.15 | 94.19 | 59.13 | 73.57 | 92.00 | 49.70 | 56.50 | 86.64 | 24.82 | 49.29 | 84.20 | 16.60 |
| ResNet34 | 78.68 | 93.84 | 58.89 | 74.26 | 92.10 | 50.30 | 58.89 | 87.44 | 27.25 | 52.18 | 84.60 | 20.60 |
| ResNet50 | 77.94 | 93.53 | 58.34 | 74.88 | 92.40 | 50.70 | 60.35 | 87.38 | 28.09 | 55.00 | 86.20 | 22.00 |
| VGG16 | 76.95 | 92.88 | 57.32 | 71.24 | 91.50 | 44.40 | 67.04 | 90.27 | 38.71 | 60.36 | 89.20 | 26.20 |
| MLPMixer/ViT | 66.69 | 87.95 | 40.93 | 60.25 | 84.50 | 33.00 | 56.97 | 85.60 | 22.42 | 51.23 | 84.20 | 17.20 |

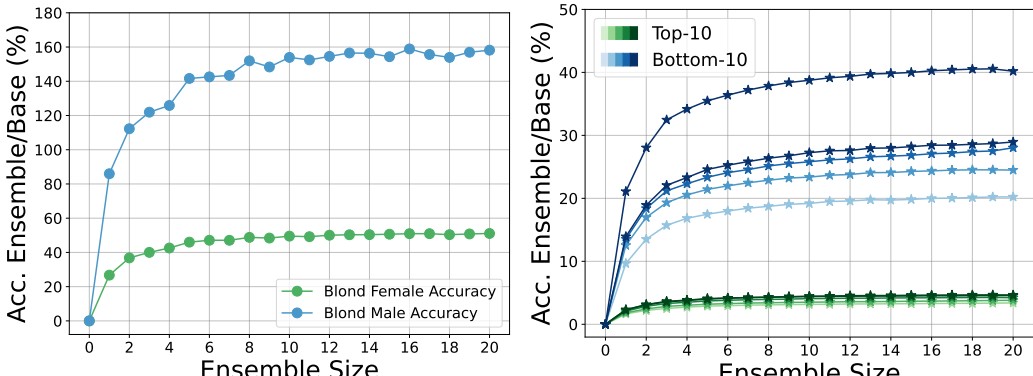

Figure 3: **Left:** CelebA test set accuracy gain as a ratio of ensemble accuracy % over the singular base model (**y-axis**) by group (majority and minority). Male is the protected attribute and Blond Males are extremely underrepresented in the training data. Nevertheless, we clearly observe that as the number of models within the homogeneous ensemble grows (**x-axis**), the protected attribute group's classification accuracy outgrows the majority group's. **Right:** CIFAR100-C test set accuracy gain ratio (same as **left** per-group (top-k and bottom-k) of the homogeneous ensemble as the number of models being aggregated increases (**x-axis**) for varying severity of corruption levels in CIFAR100-C (recall Fig. 20) from light to dark color shading. A striking observation is that not only does homogeneous DNN ensembling improves fairness by increasing the performance on the bottom group more drastically than on the top group, this effect is even more prominent at higher corruption severity levels.

k classes against a gain of around 4% for the top-k group across settings. As a result, we obtain that *even when ensembling models that share all their hyperparameters, data, and training settings, fairness naturally emerges*. Given these observations, one may wonder how does the number of models in the homogeneous ensemble impact fairness benefits. In Fig. 2 and Fig. 8, we plot fairness impact as a function of $m$, the number of models being used. A key observation we obtain is that *while the top-k group's performance plateaus rapidly for small $m$, the bottom-k group still exhibits improvements when reaching $m = 20$. We further explore increases of $m$ in the Appendix, where we consider up to 50 model ensembles (see Fig. 17). In both TinyImageNet and CIFAR100 datasets, the absolute accuracy improvements of architectures such as ResNet9, ResNet50, and VGG16 all slowly plateaued as $m \to 50$; we also present the relative test set accuracies in Figs. 18 and 19.

**Controlled Experiment: CelebA** Beyond looking at the top-k and bottom-k classes, we leverage the CelebA dataset which contains fine-grained attributes to study the fairness impact of homogeneous ensembles. Using the ResNet18 architecture, we train 20 models and measure their performances on the protected **gender:male** attribute. Employing homogeneous ensembles, we observe the average performance for the Blonde classification task to increase from 92.02% to 94.04%. Furthermore, for the protected gender attribute, we see the average performance increase from 9.44% to 21.80%, a considerable benefit that alleviates the disparate impact on an under-represented attribute. As we previously observed, homogeneous ensembles provide a disproportionate accuracy gain in the minority subgroup as further depicted in Figs. 3 and 7.

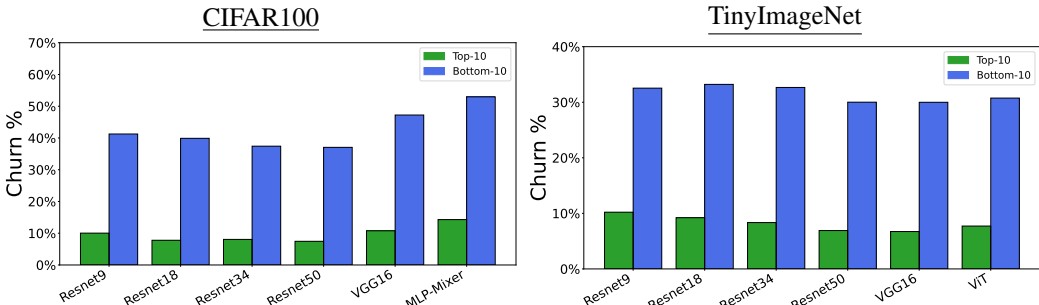

Figure 4: Depiction of churn results across models and datasets. The results demonstrate that churn is significantly higher for the bottom-k group compared to the top-k group, indicating that ensembling these models disproportionately impacts the bottom-k group (as defined in Eq. (1)). The difference in churn between bottom-k and top-k groups varies based on model architecture, suggesting that some homogeneous ensembles achieve more fairness than others.

**Controlled Experiment: CIFAR100-C** (Hendrycks & Dietterich, 2018) is an artificially constructed dataset of 19 individual corruptions on the CIFAR100 Test Dataset as depicted in Fig. 20, each with a severity level ranging from 1 to 5. Our goal is to understand the relation between fairness benefits for the bottom-k group and severity of the input corruption. We thus propose to benchmark our homogeneous ensembles on all severity levels, and for completeness, we benchmark and average performance across all corruptions for each severity level. In Fig. 3, we depict the gain in test-set accuracy achieved by the top-k and bottom-k (K=10) classes as the ensemble size ($m$) increases *relative* to a single model. We see that, consistent with earlier results, gains on top-k plateau earlier as the size of the ensemble increases. However, *the benefits of homogeneous ensembles are even more pronounced when the data is increasingly corrupted*. We observe in Fig. 21 that the largest fairness benefits occur with the maximum severity, with a maximum relative gain of 40.17% for severity 5 vs 20.18% for severity 1.

## 4 WHY HOMOGENEOUS ENSEMBLES IMPROVE FAIRNESS

We established in the previous Sec. 3 that homogeneous ensembles overly benefit minority subgroup performance. However, it is still unclear why. In this section, we take a step towards understanding that effect through the scope of model disagreement, and in particular how the only three sources of stochasticity in homogeneous ensemble may impact those results.

### 4.1 DIFFERENCE IN CHURN BETWEEN MODELS EXPLAINS ENSEMBLE FAIRNESS

It might not be clear a priori how to explain the disparate impact of homogeneous deep ensembling in bottom-k groups compared to top-k groups, as we observed in the previous Sec. 3, however we do know that such benefit only appears if the individual models do not all predict the same class, i.e., there is disagreement between models. One popular metric of model disagreement known as the *churn* will provide us with an obvious yet quantifiable answer.

**Experiment set-up.** To understand the benefit of model ensembling one has to recall that if all the models within the ensemble agree, then there will not be any benefit to aggregating the individual predictions. Hence, model disagreement is a key metric that will explain the stark change in performance that our homogeneous DNN ensembles have shown on the bottom-k group. We consider differences in churn between top-k and bottom-k. We also recall that the predictive churn is a measure of predictive divergence between two models. There are several different proposed definitions of predictive churn (Chen et al., 2020; Shamir & Coviello, 2020; Snapp & Shamir, 2021); we will employ the one that is defined on two models $f_1$ and $f_2$ as done by (Milani Fard et al., 2016) as the fraction of test examples for which the two models disagree:

$$C(f_1, f_2) = \mathbb{E}_{\mathcal{X}}\left[\mathbb{1}_{\{\hat{y}_{x;f_1} \neq \hat{y}_{x;f_2}\}}\right], \tag{1}$$

where $\mathbb{1}$ is the indicator. For an ensemble with more than two models, we will report the average churn computed across 100 randomly sampled (without replacement) pairs of models. As a further motivation to employ Eq. (1), we provide in Fig. 22 the strong correlation between Churn(%) and Test accuracy improvement(%) for various architectures on both CIFAR100 and TinyImageNet. In

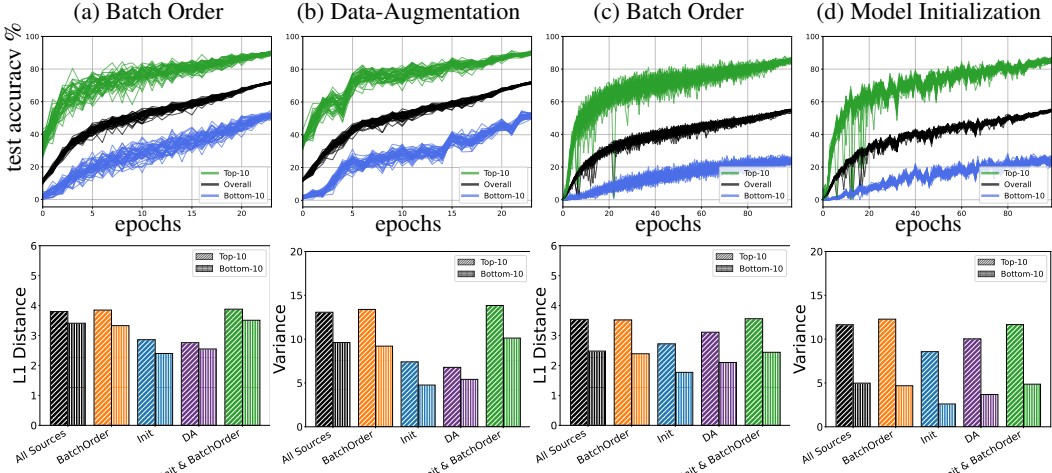

Figure 5: Depiction of multiple individual training episodes of a ResNet9 model on CIFAR100 (**two left columns**) and ResNet50 model on TinyImageNet (**two right columns**). We clearly observe that varying one factor of stochasticity at a time highlights which ones provide the most randomness between training episodes. In this setting, we see that batch ordering is the main source. On the other hand, model-init and data-augmentation have little effect and we even observe very similar trends at different epochs between the individual runs.

fact, the Pearson correlation coefficient (a maximum score of 1 indicates perfect positive correlation) between churn and test set accuracy are $0.975$ for CIFAR100 and $0.93$ for TinyImageNet i.e., a greater value for Eq. (1) is an informative proxy on the impact toward test set accuracy.

**Observations.** In Fig. 4, we report churn for various architectures on CIFAR100 and TinyImagenet. We observe that architectures differ in the overall level of churn, but a consistent observation across architectures emerges: there are large gaps in the level of churn between top-k and bottom-k. For example, on ResNet18 for TinyImageNet the difference is churn of $9.22\%$ and $33.21\%$ for top-k and bottom-k respectively, while it is $7.78\%$ and $39.89\%$ for top-k and bottom-k for CIFAR100. In short, the models disagree much more when looking at samples belonging to the bottom-k groups than when looking at samples belonging to the top-k groups. In fact, when looking at the samples of the bottom-k classes, the models vary in which samples are incorrectly classified (by definition of churn, please see Eq. (1)). As a result, that group benefits much more from homogeneous ensembling.

From these observations, it becomes clear that poor performance from individual models on the bottom-k subgroups does not stem from a systematic failure and can thus be overcome through homogeneous ensembling.

## 4.2 CHARACTERIZING STOCHASTICITY IN DEEP NEURAL NETWORKS TRAINING

While Sec. 3 demonstrated the fairness benefits of homogeneous ensembles, and Sec. 4.1 linked those improvements to increased disagreement between the individual models for the minority group and bottom-k classes, one question remains unanswered: what drives models trained with the same hyperparameters, optimizers, architectures, and training data to end-up disagreeing? This is what we propose to answer in this section by controlling each of the possible sources of randomness that impact training of the individual models.

To understand more what introduces the most significant levels of stochasticity, we first explore how different sources of randomness impact the training trajectories of DNNs. In particular, for homogeneous ensembles there are only three source of randomness: (i) *Random Initialization* (Glorot & Bengio, 2010; He et al., 2016b), (ii) *Data augmentation* realizations (Kukačka et al., 2017; Hernández-García & König, 2018), and (iii) *Data shuffling and ordering* (Smith et al., 2018; Shumailov et al., 2021). Clearly, if a source introduces low randomness, different training episodes will produce models with low disagreement and thus low fairness benefits. To identify the impact of the above three sources separately, we perform a thorough ablation study.

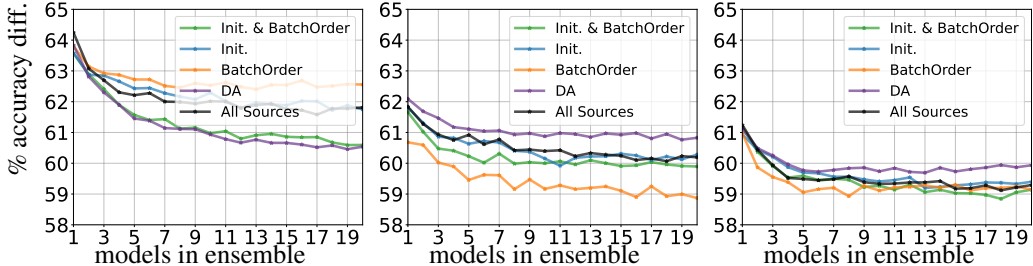

Figure 6: Accuracy % difference between top and bottom 10 classes, for ResNet18 (**left**), 34 (**middle**), and 50 (**right**) for TinyImageNet. We clearly observe that once we control for the different sources of stochasticity, it is possible to skew the ensemble to favor the bottom group, in which case fairness is further amplified compared to the baseline ensemble. Although the trends seem mostly consistent across architectures of the same family (ResNets) and datasets, it is not necessarily the case between architecture families: see Fig. 13 for ResNet-CIFAR100, Fig. 15 for MLPMixer/VGG16-CIFAR100 andViT/VGG16-TinyImagenet.

**Experiment set-up.** To isolate the impact of the different sources of stochasticity, we propose an ablation study of the following sources: *Change Model Initialization* (`Init`): for this ablation, we change the model initialization weights by changing the torch seed for each model before the model is instantiated. *Change Batch Ordering* (`BatchOrder`): for this ablation, we change the ordering of image data in each mini-batch by changing the seed for the dataloader for each model training. *Change Model Initialization and Batch Ordering* (`Init & BatchOrder`): for this ablation, both the model initialization and batch ordering are changed for each model training. *Change Data Augmentation* (`DA`): for this ablation, only the randomness in the data augmentation (e.g. probability of random flips, probability of CutMix(Yun et al., 2019), etc.) is changed. The relevant torch and numpy seeds are changed right before instantiating the data augmentation pipeline. Custom fixed-seed data augmentations is also used. *Change Model Initialization, Batch Ordering and Data Augmentation* (`All Sources`): for this ablation, the model initialization, batch ordering and data augmentation seeds are changed for each model training–this ablation represents the standard homogeneous ensemble of Sec. 3. A last source of randomness can emerge from hardware or software choices and round-off errors (Zhuang et al., 2022; Shallue et al., 2019) which we found to be negligible compared to the others. In addition to providing training curves evolution for each ablation, we also use two quantitative metrics. First, we will leverage the `L1-Distance` of the accuracy trajectories during training, which is calculated for every epoch by averaging the absolute distance in accuracy among the ensemble members and averaging these values across the training epochs. Second, we will leverage the `Variance` of the different training episodes' accuracy at each epoch and then average over all the epochs.

**Observations.** In Fig. 5, we plot these measures of stochasticity for both CIFAR100 and TinyImageNet on different DNNs. We observe that the single sources of noise dominate, such that the ablations themselves equate to the level of noise in the DNN with all sources of noise present. In particular, we observe one striking phenomenon: the variation of the data ordering within each epoch between training trajectories `BatchOrder` is the main source of randomness. It is equivalent to the level of noise we observe for the DNN with all sources of noise `All Sources`, and the DNN with the ablation `Init & BatchOrder`. As seen in Fig. 5 when the batch ordering is kept the same across training episodes, varying the data-augmentation and/or the model initialization has very little impact.

### 4.3 CAN DIFFERENT SOURCES OF STOCHASTICITY IMPROVE HOMOGENEOUS DEEP ENSEMBLE FAIRNESS?

The last important point that needs to be addressed is to relate the amount of randomness that each of the three sources introduce (recall Sec. 4.2) with the actual fairness benefits of the homogeneous ensemble. In fact, Fig. 5 did emphasize how each source of randomness provides different training dynamics and levels of disagreements, which are the cause of the final fairness outcomes.

In Fig. 6, we depict the **accuracy difference** between average top-k and bottom-k. A value of $0$ indicates that the model performs equally on both top-k and bottom-k classes. We observe that for the majority of dataset/architecture combinations, batch ordering minimizes the gap between top and bottom-k class accuracy. Surprisingly, the resulting fairness level is even greater than when

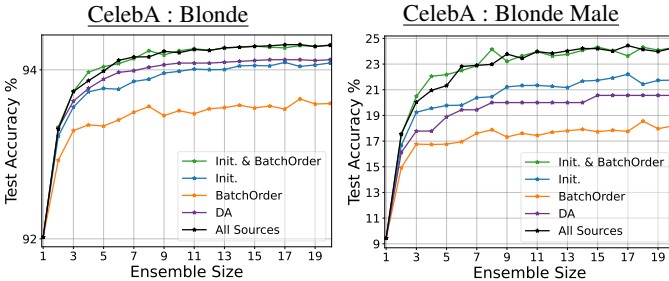

Figure 7: 20 Model Ensemble Performance (ResNet18) on Blond (**left**) and Blond Male (**right**) Classification in CelebA. In the overall blond classification, the max accuracy increased up to 2.27% using homogeneous ensembling, whereas for the Blond Male minority group the max accuracy increased dramatically–up to 14.99%–in the same ensembles.

employing all the source of stochasticity, i.e., *it is possible to further improve the emergence of fairness in homogeneous ensembles solely by varying the batch ordering between the individual models*. In Fig. 7, we observe that although gains quickly plateau for the Blonde category in all sources, the stochasticity introduced by initialization and batch ordering `Init & BatchOrder` matches, and sometimes outperforms the noise ablation on the minority group performance. There is one exception to this, as we see that data-augmentation variation for ResNet18 on TinyImageNet creates the largest decrease. This observation is aligned with prior studies which compared the variability of a learned representation as a function of the different sources of stochasticity present during training.

In Appendix B. of (Fort et al., 2019), the authors note that at higher learning rates, mini-batch shuffling adds more randomness than model initialization due to gradient noise. Since our experiments for CIFAR-100 and TinyImageNet use higher learning rates, this is in line with the observations from (Fort et al., 2019). Additionally, we also perform an ablation on learning rates Fig. 24 in Appendix H where one can clearly see the impact of different hyperparameters onto the final conclusions. There are also several works to-date that have considered how stochasticity can impact top-line metrics (Nagarajan et al., 2018). Most relevant to our work is (Qian et al., 2021; Zhuang et al., 2022; Madhyastha & Jain, 2019; Summers & Dinneen, 2021) that evaluates how stochasticity in training impacts fairness in DNN Systems. However, all the existing works have restricted their treatment to a single model setting, and do not evaluate the impact of ensembling.

## 5 CONCLUSION AND FUTURE WORK

In this work, we establish that while ensembling DNNs is often seen as a method of improving average performance, it can also provide significant fairness gains–even when the apparent diversity of the individual models is limited, e.g., only varying through the batch ordering or parameter initialization. This suggests that homogeneous ensembles are a powerful tool to improve fairness outcomes in sensitive domains where human welfare is at risk, as long as the number of employed models is pushed further even after the average performance plateaus (recall Sec. 3). Our observations led us to precisely understand the cause for the fairness emergence. In short, by controlling the different sources of randomness, we were not only able to measure the impact of each source onto the final ensemble diversity, but we were also able to pinpoint initialization and batch ordering as the main source of diversity. We hope that our observations will open the door to address fairness in homogeneous ensemble through a principled and carefully designed control of the sources of stochasticity in DNN training.

**Limitations: Validity on non-DNN models/non-image datasets.** While our study focuses on image datasets and DNNs, we found that the fairness benefits of homogeneous ensembles extend beyond such settings. For example, we have conducted additional experiments on the Adult Census Income dataset(Becker & Kohavi, 1996) using both a 3-layer multi-layer perceptron (MLP) model and a Decision Trees model. In the MLP setting, we used both race and sex as sensitive attributes. We trained on the remaining 12 features to predict income level >$50k. Using the same ablations as our DNN experiments we report in Tab. 2: homogeneous ensembling improves the >$50k Amer-Indian-Eskimo subgroup prediction performance by 2.53%. As for the Decision Trees, we limited the max depth to be 10 and used the random state, which affects the random feature permutation, as the source of stochasticity to control. The results, shown in Tab. 3, depict improved fairness for Black and Amer-Indian-Eskimo. The >$50k Other races subgroup had an outsized improvement with a 3.84% increase in accuracy over the base model. This motivates the need to develop novel theories explaining the fairness benefits of homogeneous ensembles, as those benefits are not limited to DNNs or image datasets.

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
