# A  APPENDIX

# B  EXPERIMENTAL SETUP

## B.1  SAMPLING

Given a pool of $M$ models, for each ensemble size $S$ we sample 100 times with replacement. We then average the accuracy across the 100 samples plus one base model that is shared across all variants. The result at each $S$ is reported until an ensemble of $M$ models is reached.

## B.2  CIFAR-100 TRAINING

We use the following architectures: ResNet9 (He et al., 2016a), VGG16 (Simonyan & Zisserman, 2014) and MLP-Mixer (Tolstikhin et al., 2021). We train them as follows:

**ResNet-9** We train the model for 24 steps using Stochastic Gradient Descent (SGD). We implemented standard data augmentation by applying Random Horizontal Flip, Random Translate, and Cutout. We use a Slanted Triangular Learning Rate (SLTR) (Howard & Ruder, 2018). The top-1 test set accuracy is 72.24%

**ResNet18/34/50** For these 3 ResNet architectures, we train the model for 50 epochs using Stochastic Gradient Descent (SGD), batch size of 512, momentum=0.9, and weight decay=0.0005. We implemented standard data augmentation by applying Random Horizontal Flip, Random Crop, Random Affine, and Cutout. We use a combination of warmup for the first 5 epoch and cosine annealing for scheduler. The top-1 test set accuracy for ResNet-18 is 73.56%, ResNet-34 is 74.24%, and ResNet-50 is 74.89%

**VGG16** We train the model for 130 epochs using Stochastic Gradient Descent (SGD). We implemented standard data augmentation by applying Random Horizontal Flip, Random Crop, and Random Rotation. We use a combination of warmup for 1 epoch and a multi-step scheduler with milestones at steps 60 and 120. The top-1 test set accuracy is 71.23%

**MLP-Mixer** We train the model for 300 steps using Adaptive Moment Estimation (Adam) (Kingma & Ba, 2014). We implemented standard data augmentation by applying Random Crop, AutoAugment (CIFAR10 Policy) (Cubuk et al., 2018), and CutMix (Yun et al., 2019). We use a combination of warmup for the first 5 epoch and cosine annealing for scheduler. The top-1 test set accuracy is 60.28%

## B.3  TINYIMAGENET TRAINING

We use the following architectures: ResNets (He et al., 2016a), VGG-16 (Simonyan & Zisserman, 2014) and ViT (Dosovitskiy et al., 2020). We train them as follows:

**ResNets** We train 3 different architectures from the ResNet family (ResNet18, 34, 50) for 100 steps using Stochastic Gradient Descent (SGD). We implemented standard data augmentation by applying Random Resized Crop and Random Horizontal Flip. We use a Slanted Triangular Learning Rate (SLTR) (Howard & Ruder, 2018). The top-1 test set accuracy for ResNet-18 is 49.27%, ResNet-34 is 52.18%, and ResNet-50 is 54.99%

**VGG16** We train the model for 100 steps using Stochastic Gradient Descent (SGD). We implemented standard data augmentation by applying Random Resized Crop and Random Horizontal Flip. We use a Slanted Triangular Learning Rate (SLTR) (Howard & Ruder, 2018). The top-1 test set accuracy is 60.37%

**ViT** We train the model for 100 steps using Adaptive Moment Estimation with decoupled weight decay (AdamW) (Loshchilov & Hutter, 2017). We implemented standard data augmentation by applying Random Horizontal Flip, Random Resized Crop, AutoAugment (Cubuk et al., 2018), Random Erasing (Zhong et al., 2020), Cutmix (Yun et al., 2019), and Mixup(Zhang et al., 2017). We use a combination of warmup for the first 10 epoch and cosine annealing (Loshchilov & Hutter, 2016) for scheduler. The top-1 test set accuracy is 51.21%

## C   FAIR-ENSEMBLE: WHEN HOMOGENEOUS ENSEMBLE DISPROPORTIONATELY BENEFIT MINORITY GROUPS

### C.1   EXPERIMENTAL SETUP

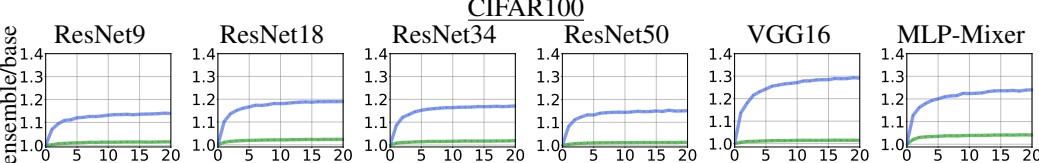

Figure 8: Depiction of the accuracy gain as a ratio of ensemble accuracy % over the singular base model (**y-axis**) by per-group (top-k and bottom-k) test set accuracies for CIFAR100.

### C.2   BALANCED DATASET SUB-GROUPS

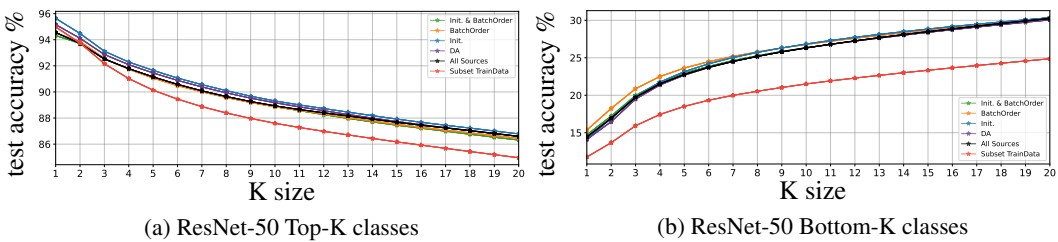

(a) ResNet-50 Top-K classes

(b) ResNet-50 Bottom-K classes

Figure 9: Average Top and Bottom-K accuracy as number of K-classes increase. We observe that **DA** and **Init.** outperforms **All Sources** baseline performance in Top-K classes, whereas **BatchOrder** and **Init.** outperfroms **All Sources** on Bottom-K classes. In both top and bottom groups, only the **TrainSubsetData** variant underperforms **All Sources**.

## C.3 CIFAR-100

Figure 10: Accuracy for Top-K and Bottom-K across models added to ensemble on CIFAR100

## C.4 TINYIMAGENET

Figure 11: Accuracy for Top-K and Bottom-K across models added to ensemble on TinyImageNet

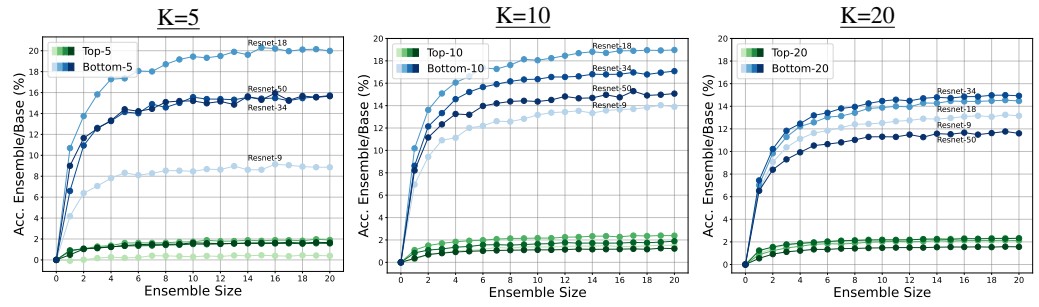

Figure 12: Top/Bottom-K performance for K = {5,10,20} in CIFAR100

# D   CONTROLLING FOR THE SOURCES OF STOCHASTICITY IN HOMOGENEOUS ENSEMBLES

## D.1   CIFAR100 ACCURACY % DIFFERENCE FOR RESNET 18, 34, 50

ResNet family on CIFAR100

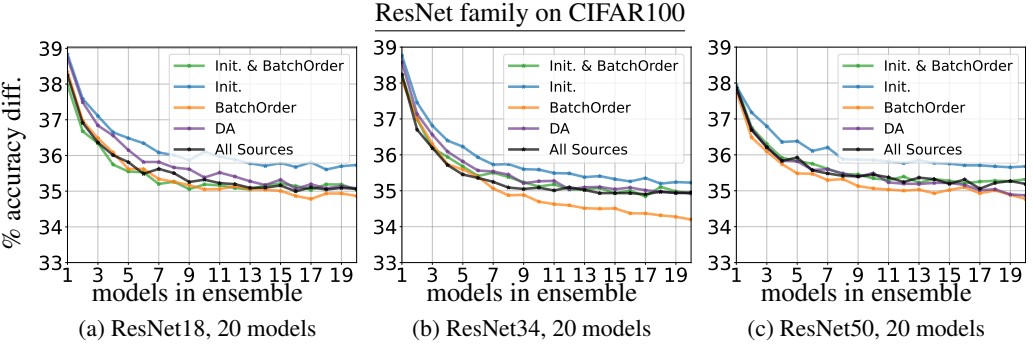

(a) ResNet18, 20 models    (b) ResNet34, 20 models    (c) ResNet50, 20 models

Figure 13: Accuracy % difference between top and bottom 10 classes, for ResNet18, 34, and 50 for CIFAR100.

## D.2   CIFAR100 AND TINYIMAGENET RESULTS

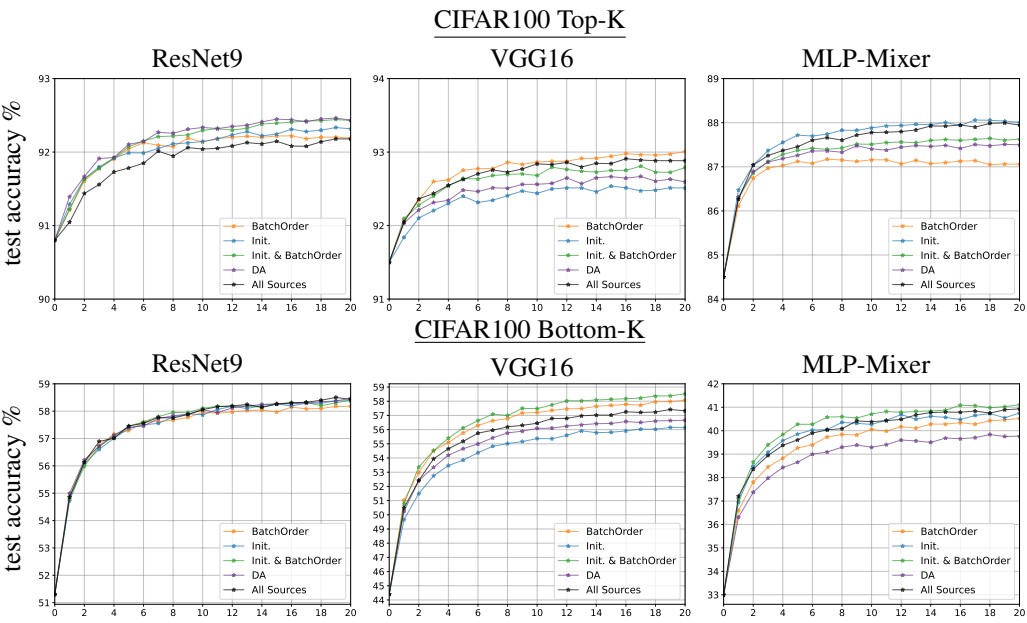

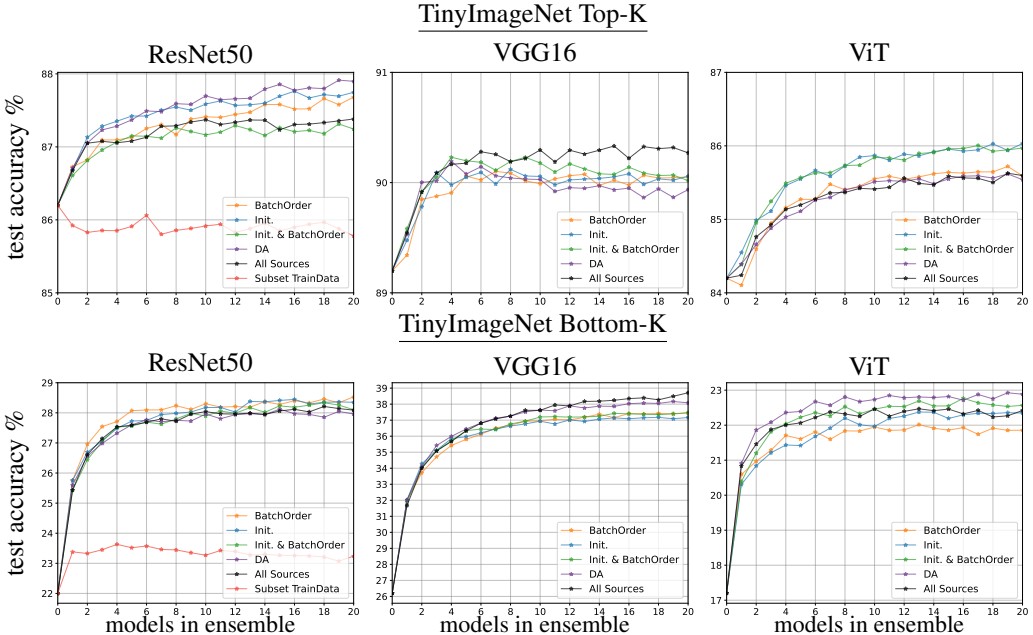

Figure 14: Average Test Accuracy on CIFAR100 and TinyImageNet for Top-K and Bottom-K ($K = 10$) Performing Classes

### D.3 DOMINANT SOURCES OF STOCHASTICITY

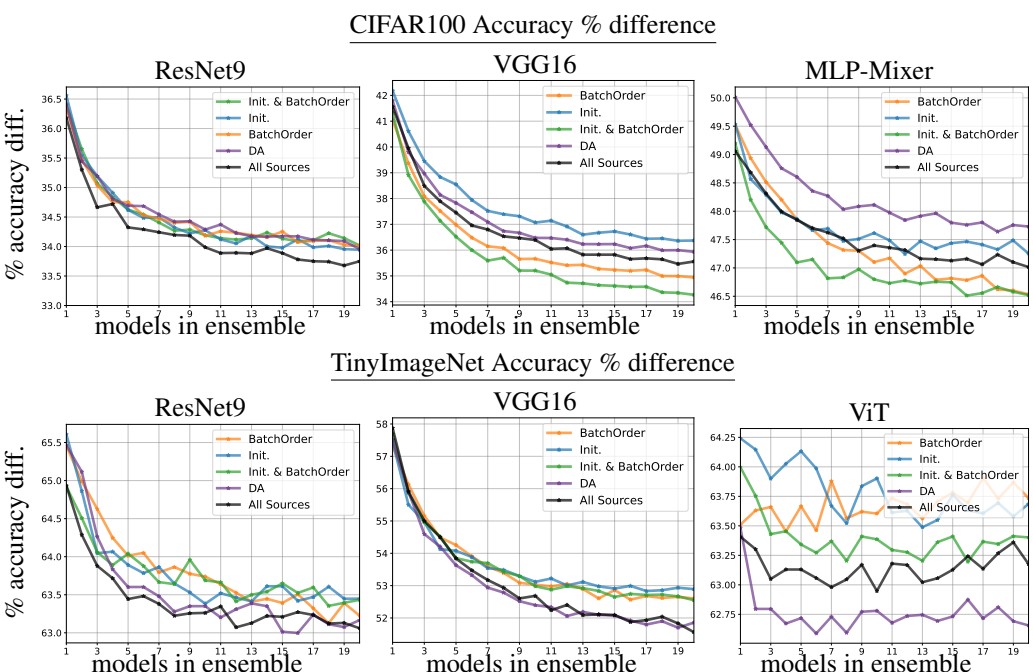

Figure 15: Accuracy % difference between top and bottom 10 classes for ResNet-9, VGG16, and MLPMixer trained on CIFAR100 and TinyImageNet

# E RESULTS AND DISCUSSION

## E.1 COMPARISON BETWEEN RESNET ARCHITECTURES

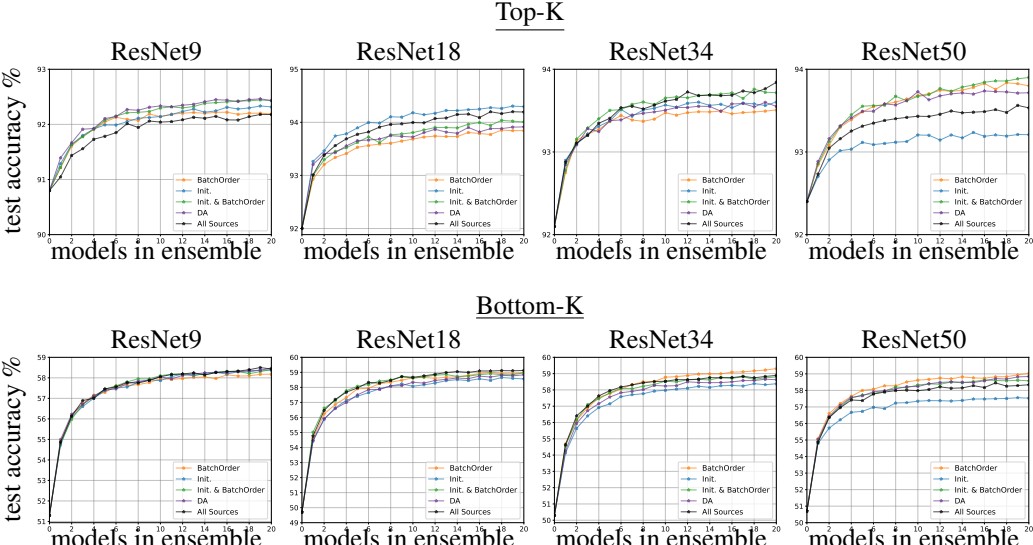

Figure 16: Average Test Accuracy on CIFAR100 for Top-K and Bottom-K classes across different sizes of ResNets.

## E.2 BENEFITS OF EVEN LARGER HOMOGENEOUS ENSEMBLES

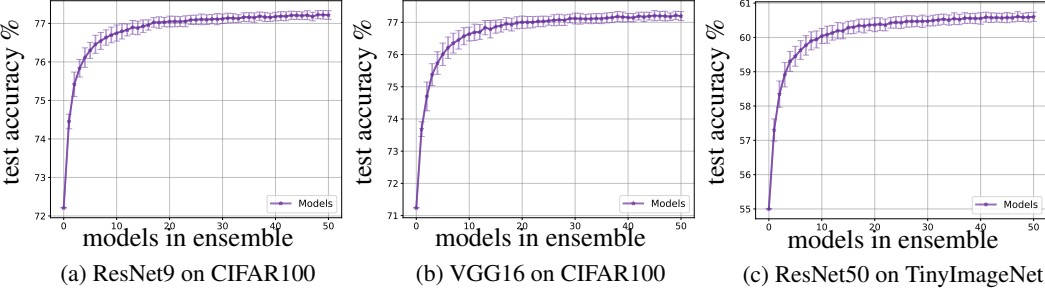

Figure 17: Average Accuracy per size of homogeneous ensemble. The average accuracy for each model added is calculated by averaging 100 random samples from a population of 50 models. We can see that the average accuracy starts to slowly plateau as the ensemble grows to 50 models.

## E.3    CIFAR-100

Figure 18: Ratio of Top & Bottom K ensemble accuracy for different model architectures and ensemble sizes on CIFAR100

### E.4 TinyImageNet

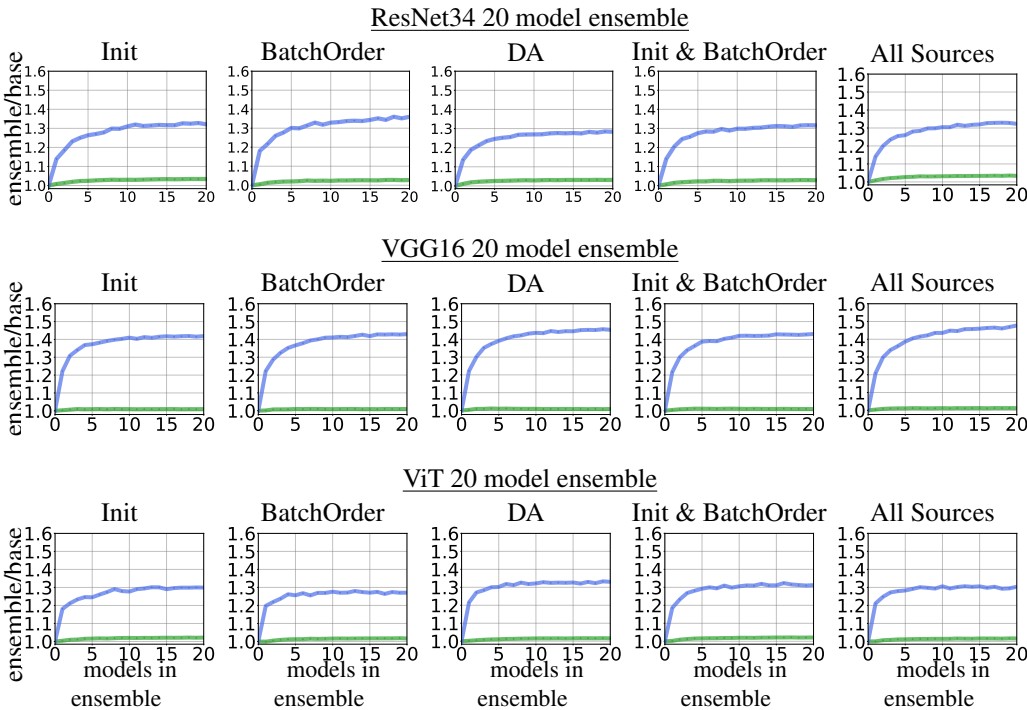

Figure 19: Ratio of Top & Bottom K ensemble accuracy for different model architectures and ensemble sizes on TinyImageNet

# F FAIR ENSEMBLE: IMPROVED ROBUSTNESS

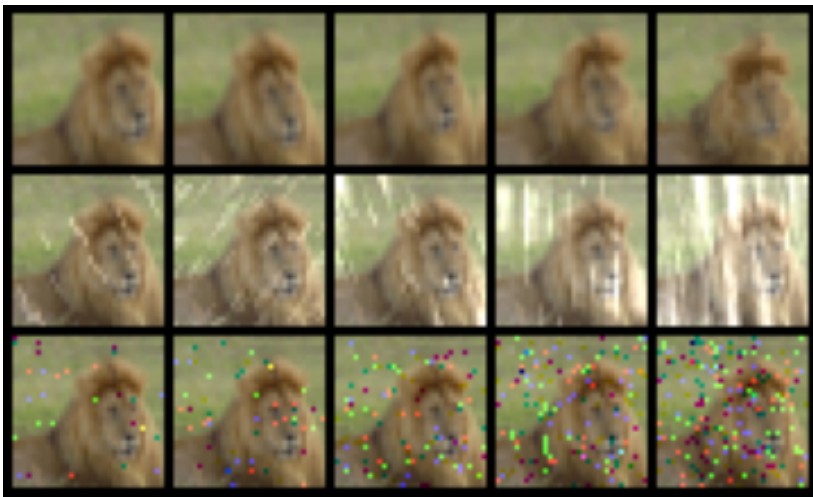

Figure 20: For each **row** we depict three of the different types of corruptions from the CIFAR100-C dataset (`elastic_transform`, `snow`, and `impulse_noise` respectively), and for each **column** we depict the corruption severity levels ($1 \mapsto 5$). The image belongs to the `lion` class.

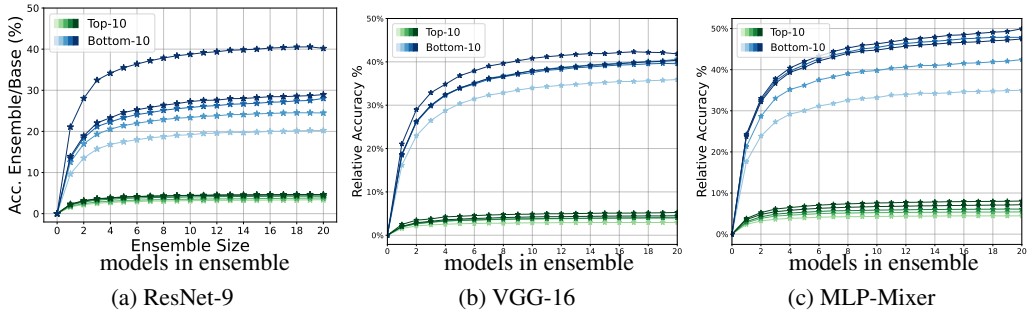

|  |  |  |
|:---:|:---:|:---:|
| (a) ResNet-9 | (b) VGG-16 | (c) MLP-Mixer |

Figure 21: Performance on CIFAR-100 Corrupt based on Severity Levels.

# G DIFFERENCE IN CHURN BETWEEN MODELS EXPLAINS ENSEMBLE FAIRNESS

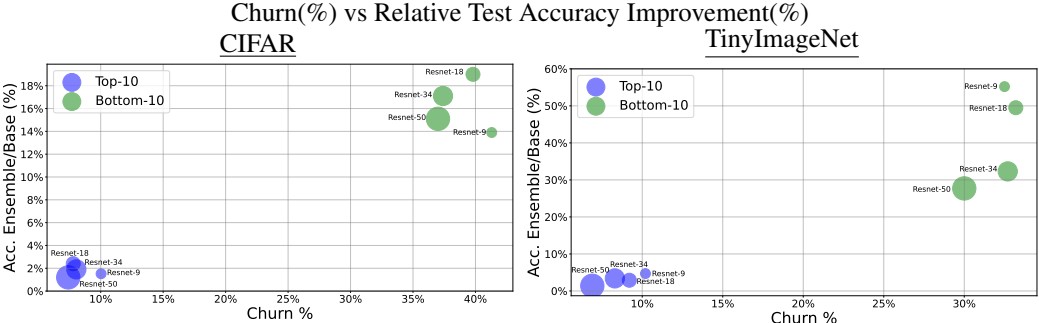

Figure 22: Correlation between Model Disagreement and Ensemble Performance

# H CAN DIFFERENT SOURCES OF STOCHASTICITY IMPROVE HOMOGENEOUS DEEP ENSEMBLE FAIRNESS?

## H.1 CONTRIBUTION OF STOCHASTICITY IN ENSEMBLES

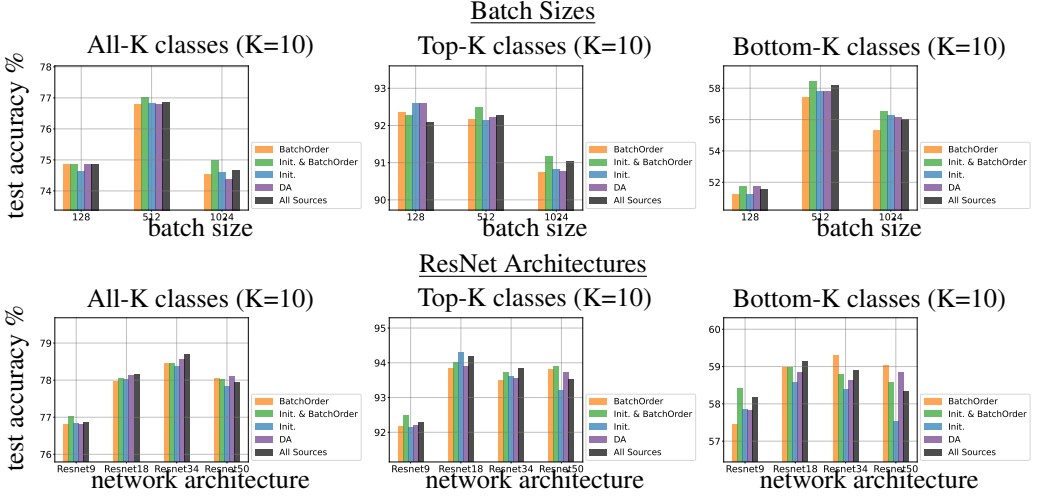

Figure 23: Average Test Accuracy on CIFAR100 as batch and architecture size increases. Batch 512 is default.

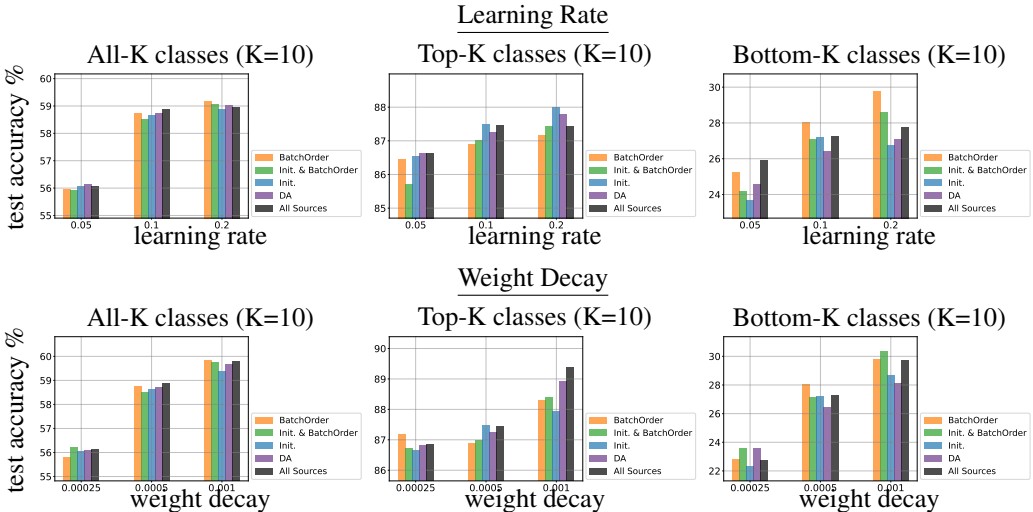

Figure 24: Average Eval Accuracy on TinyImageNet as learning rate and weight decay increases. 0.1 is default learning rate, and 0.0005 is default weight decay.

# I   HOMOGENEOUS ENSEMBLES IN NON-DNN MODELS/NON-IMAGE DATASETS

Table 2: MLP ensemble performance over Adult Census Income subgroups with sensitive attributes

|  | Base Model | 10-model ensemble | | | |
|---|---|---|---|---|---|
|  |  | BatchOrder | Initialization | Init & BatchOrder | All Sources |
| >$50k | **79.93** | $79.87 \pm 0.03$ | $79.75 \pm 0.28$ | $79.57 \pm 0.38$ | $79.68 \pm 0.21$ |
| >$50k Male | 25.98 | $26.75 \pm 0.19$ | $26.1 \pm 0.49$ | $\mathbf{27.17 \pm 0.83}$ | $26.96 \pm 0.51$ |
| >$50k Female | 27.97 | $28.63 \pm 0.21$ | $28.23 \pm 0.41$ | $\mathbf{28.93 \pm 0.81}$ | $28.76 \pm 0.4$ |
| >$50k White | 26.48 | $27.22 \pm 0.17$ | $26.64 \pm 0.47$ | $\mathbf{27.61 \pm 0.81}$ | $27.4 \pm 0.48$ |
| >$50k Nonwhite | 24.44 | $25.2 \pm 0.45$ | $24.25 \pm 0.6$ | $\mathbf{25.83 \pm 0.88}$ | $25.62 \pm 0.62$ |
| >$50k Black | 24.02 | $25.29 \pm 0.67$ | $23.92 \pm 0.74$ | $\mathbf{25.89 \pm 0.66}$ | $25.73 \pm 0.57$ |
| >$50k Asian-Pac-Islander | 23.31 | $23.64 \pm 0.39$ | $22.82 \pm 0.52$ | $\mathbf{24.14 \pm 0.96}$ | $23.98 \pm 0.7$ |
| >$50k Amer-Indian-Eskimo | 26.32 | $26.32 \pm 0$ | $27.79 \pm 3.5$ | $\mathbf{28.85 \pm 4.73}$ | $27.74 \pm 3.14$ |
| >$50k Other | 32 | $32 \pm 0$ | $31.44 \pm 1.39$ | $\mathbf{32.12 \pm 0.68}$ | $32.04 \pm 0.4$ |

Table 3: Decision Trees ensemble performance over Adult Census Income subgroups with sensitive attributes

|  | Base Model | 10-Model Ensemble |
|---|---|---|
| >$50k | 85.85 | $\mathbf{85.91 \pm 0.02}$ |
| >$50k Male | 60.26 | $\mathbf{60.35 \pm 0.04}$ |
| >$50k Female | 55.59 | $\mathbf{56.04 \pm 0.09}$ |
| >$50k White | 59.89 | $\mathbf{59.99 \pm 0.05}$ |
| >$50k Nonwhite | 56.18 | $\mathbf{56.73 \pm 0.05}$ |
| >$50k Black | **52.51** | $\mathbf{52.51 \pm 0}$ |
| >$50k Asian-Pac-Islander | 61.65 | $\mathbf{62.41 \pm 0}$ |
| >$50k Amer-Indian-Eskimo | **63.16** | $\mathbf{63.16 \pm 0}$ |
| >$50k Other | 48 | $\mathbf{51.84 \pm 0.78}$ |

## J  TOP AND BOTTOM CLASSES FOR CIFAR100 AND TINYIMAGENET

### J.1  CIFAR100

Table 4: Top-10 and Bottom-10 class names for CIFAR100. The classes are from the averaged test accuracies from the 20-model ensembles.

| ResNet9 | ResNet18 | ResNet34 | ResNet50 | VGG16 | MLP-Mixer |
|---|---|---|---|---|---|
| Top-10 | | | | | |
| wardrobe | skunk | skunk | orange | road | wardrobe |
| motorcycle | orange | road | wardrobe | wardrobe | motorcycle |
| orange | motorcycle | orange | motorcycle | sunflower | orange |
| skunk | road | sunflower | skunk | motorcycle | sunflower |
| road | wardrobe | motorcycle | road | skyscraper | road |
| chimpanzee | palm_tree | wardrobe | sunflower | skunk | skyscraper |
| sunflower | chimpanzee | palm_tree | chimpanzee | palm_tree | keyboard |
| orchid | sunflower | pickup_truck | palm_tree | orange | palm_tree |
| mountain | tractor | aquarium_fish | aquarium_fish | chair | plain |
| apple | skyscraper | skyscraper | lawn_mower | chimpanzee | skunk |
| Bottom-10 | | | | | |
| man | mouse | shark | girl | possum | mouse |
| shark | bear | possum | lizard | crocodile | bowl |
| lizard | shark | crocodile | possum | girl | woman |
| bowl | girl | lizard | maple_tree | shark | girl |
| possum | lizard | girl | bear | bear | squirrel |
| shrew | man | man | otter | lizard | possum |
| seal | otter | bowl | bowl | seal | lizard |
| girl | seal | otter | man | boy | boy |
| otter | bowl | seal | boy | otter | otter |
| boy | boy | boy | seal | man | seal |

### J.2  TINYIMAGENET

Table 5: Top-10 and Bottom-10 wnid names for TinyImageNet. The names are from the averaged test accuracies from the 20-model ensembles.

| ResNet9 | ResNet18 | ResNet34 | ResNet50 | VGG16 | ViT |
|---|---|---|---|---|---|
| Top-10 | | | | | |
| n02791270 | n02791270 | n02791270 | n02791270 | n02791270 | n07875152 |
| n02509815 | n02509815 | n02509815 | n02509815 | n03042490 | n03814639 |
| n03976657 | n02906734 | n02906734 | n02906734 | n02509815 | n03983396 |
| n02124075 | n03042490 | n03814639 | n03042490 | n03814639 | n03042490 |
| n03814639 | n03814639 | n01950731 | n01950731 | n02906734 | n02823428 |
| n03089624 | n03976657 | n03599486 | n04067472 | n01950731 | n03599486 |
| n03983396 | n01950731 | n03042490 | n03599486 | n04398044 | n02509815 |
| n02002724 | n04560804 | n03976657 | n03976657 | n02124075 | n02791270 |
| n03126707 | n03599486 | n04067472 | n07579787 | n03089624 | n03126707 |
| n03447447 | n02002724 | n03126707 | n03126707 | n04067472 | n02906734 |
| Bottom-10 | | | | | |
| n02437312 | n04532670 | n03160309 | n03544143 | n02085620 | n02927161 |
| n04070727 | n03544143 | n01945685 | n03617480 | n04417672 | n03544143 |
| n02268443 | n04486054 | n04417672 | n04070727 | n02268443 | n04070727 |
| n01945685 | n02268443 | n04532670 | n03804744 | n04486054 | n01641577 |
| n02226429 | n03160309 | n03617480 | n03160309 | n01945685 | n02094433 |
| n02233338 | n03617480 | n01855672 | n01945685 | n02094433 | n02480495 |
| n02480495 | n01855672 | n03804744 | n02268443 | n04070727 | n02410509 |
| n02410509 | n02480495 | n02480495 | n02480495 | n02480495 | n04532670 |
| n03617480 | n02123394 | n02123394 | n02123394 | n02410509 | n02950826 |
| n02123394 | n02410509 | n02410509 | n02410509 | n02123394 | n02123394 |