# OpenReview forum: "FAIR-Ensemble: Homogeneous Deep Ensembling Naturally Attenuates Disparate Group Performances"
_ICLR.cc/2024/Conference — ICLR 2024 Conference Withdrawn Submission_

### Official Review · Reviewer_6SCz · 2023-10-29

**Soundness:** 3 good
**Presentation:** 3 good
**Contribution:** 3 good
**Rating:** 6
**Confidence:** 4

**Summary:**

This paper shows that model ensembles improve fairness, i.e. they improve performance on hard subgroups more than they do for easy subgroups. The paper provides some empirical analysis for why homogeneous ensembles can improve fairness, by measuring cross-model agreement (churn) for each split and an ablation on the three sources of randomness during training.

**Strengths:**

This paper starts with an interesting empirical observation: homogeneous ensembles disproportionately benefit harder groups (classes or subpopulations).
- (minor) it would be good to at least informally define top-k and bottom-k in figure 1 so that it is easily skimmable.

The finding seems to be robust to architecture and setting choice. It seems like a simple and fundamental property of (at least) deep neural networks, which has many implications for fairness and bias. Section 4 provides several analysis experiments which help us begin to understand this phenomenon.

**Weaknesses:**

I don't think section 4 sufficiently explains why homogeneous ensembles improve fairness. High disagreement (churn) for harder subgroups is unsurprising, given that we have already observed higher ensemble performance. The ablations for sources of randomness show some qualitative differences, but I don't think these differences _explain_ why fairness is increasing more than average or easy-group performance. After section 4, are we closer to understanding why initialization and/or batch order causes fairer ensemble performance?

I am also somewhat concerned about using the accuracy ratio to compare easy vs hard subgroups since ratios of lower numbers will naturally seem more extreme given the same gap. Maybe another way of visualizing these results is as an easy group acc vs hard group acc scatterplot with a probit transform, as done in https://proceedings.mlr.press/v139/miller21b/miller21b.pdf?

I'm unsure if "fairness" accurately describes what's going on in this paper. It is accurate for describing the results for CelebA, but it feels like a strange phrase for describing harder classes in e.g., CIFAR100.

(minor) figures 5, 6 were confusing because they have the same labels and the difference is only mentioned in the caption.

**Questions:**

How do you think non-homogeneous ensembles (e.g. different architectures, data, or training procedures) fit into the findings here? Will they show similar tendencies but just with bigger performance increase due to the diversity?

---

### Official Review · Reviewer_du9t · 2023-10-30

**Soundness:** 2 fair
**Presentation:** 2 fair
**Contribution:** 2 fair
**Rating:** 3
**Confidence:** 4

**Summary:**

The authors have discovered that utilizing a homogeneous ensemble alleviates the inconsistent performance observed when compared to a single model. They conducted some experiments and performed some analyses.

**Strengths:**

1. The problem of model fairness is both practical and deserving of attention.

2. The writing is clear and easy to follow.

**Weaknesses:**

1. The primary finding of the paper is not surprising. It's widely acknowledged that using ensembles can boost model performance. Hence, it's not unexpected that primary groups with better performance exhibit a smaller improvement due to the "marginal effect" compared to lesser-performing subgroups.
2. The authors neither proposed advanced methods nor provided any theoretical foundation for the analysis of the experimental phenomena.
3. There has been some work on addressing data heterogeneity and model fairness using ensemble techniques, but the authors overlooked them [a,b,c].

[a] Towards Understanding the Mixture-of-Experts Layer in Deep Learning, NeurIPS 2022.

[b] Long-tailed Recognition by Routing Diverse Distribution-Aware Experts, ICLR 2021.

[c] Self-Supervised Aggregation of Diverse Experts for Test-Agnostic Long-Tailed Recognition. NeurIPS 2022.

**Questions:**

See above

---

### Official Review · Reviewer_Y5JH · 2023-10-31

**Soundness:** 2 fair
**Presentation:** 2 fair
**Contribution:** 2 fair
**Rating:** 3
**Confidence:** 3

**Summary:**

This work shows that for a homogeneous ensemble of deep neural networks, where each model shares the same hyperparameters, architecture, and dataset, the accuracy on minority groups and tail classes increases disproportionately with the number of models used in the ensemble. Extensive experiments and ablation studies are performed, and the authors offer possible explanations for the phenomenon.

**Strengths:**

1. The experiments are comprehensive and cover a wide variety of difficult datasets and relevant architectures. The ablation studies are also well done.
2. The natural emergence of fairness as the number of models in the ensemble increases is an interesting observation, and the model disagreement hypothesis seems promising as an explanation.
3. The paper is well-organized, and the subheadings for observations/controlled experiments/limitations make it easy to read.

**Weaknesses:**

The weaknesses of this paper can be organized into (a) lack of specification in the training and ensembling procedure, (b) lack of contextualization and comparison to the literature, and (c) need for further investigation based on relevant literature.

1. Lack of specification in the training and ensembling procedure
    * How is the ensembling performed? Appendix B states that the ensemble accuracy is the average of the accuracies of each model, but this is contrary to the standard definition of an ensemble where the outputs of each model are combined to return a single final prediction (e.g., by averaging the logits). In particular, the accuracy of the combined prediction can be very different from the average accuracy of the individual predictions. Moreover, if the ensemble accuracy is the average of the individual model accuracies, shouldn’t the ensemble accuracy converge to the mean accuracy of all models instead of increasing monotonically with each additional model?
    * The training procedure for CelebA is not specified in Appendix B.
    * What type of pretraining is used for the models? The standard in the literature, at least for CelebA and CIFAR-100, is to pretrain on ImageNet and finetune the weights. If no pretraining is used, this may reduce the impact of the contribution, as the results may not apply to relevant settings in practice and in the literature.
    * In Figure 7, why is the test accuracy on the CelebA “blond male” group so low? It appears that a single ResNet18 achieves about 9.5% test accuracy on this group. Using ImageNet-pretrained ResNet50, a standard ERM baseline achieves about 41% worst-group test accuracy [1] while simple data balancing can improve this to nearly 80% [2].
2. Lack of contextualization and comparison to the literature
    * A critical limitation of homogeneous deep ensembles is the computation necessary to train a quantity of networks past the point at which ensemble average accuracy plateaus. I was surprised to see that the authors did not address this limitation, and I believe an in-depth discussion of its implications is important to evaluate the contribution. In particular, a comparison to the literature is necessary to contextualize the relative increase in accuracy vs. the computation required - for example, last-layer retraining [3,4] can improve worst-group accuracy on CelebA by up to 75% relative to ERM with negligible computation (47.2% to 88.0% in [3]).
    * Moreover, while it seems surprising that fairness emerges naturally in homogeneous deep ensembles, it may be a particularly inefficient method relative to explicitly learning a diverse ensemble [5]. What are the advantages of a homogeneous ensemble in this scenario?
    * I believe there is at least one important reference which should be discussed in this work [5]. In particular, note that their method avoids the computation issue by training separate heads (rather than entire models) and uses disagreement to select a single final predictor.
3. Need for further investigation based on relevant literature
    * I appreciate the effort made by the authors in Section 4 and I believe the analysis is going in the right direction. With that said, the ablation studies in Sections 4.2 and 4.3 do not necessarily answer the question posed in Section 4.1 (i.e., why the churn is significantly higher for the bottom-k group than the top-k group). I believe that this question merits additional investigation based on recent advances in the literature. One potential explanation might come from recent work into the study of loss landscapes in deep ensembles [6,7]. Are churn rates disproportionately affected by models which end up in different basins? Can these insights be used to improve the efficiency of the method, e.g., by only adding models to the ensemble which do not belong to the same basin as an existing model?
    * Another direction in the literature which may be relevant is that of ensemble calibration. In particular, [8] showed that generalization error may be estimated by the disagreement of two models trained with independent noise during SGD (similar to churn in this work), and this phenomenon is implied by the calibration of the ensemble [9]. Therefore, it may be beneficial to study how the calibration of the ensemble improves with more models added, as well as how the behavior of the calibration on the bottom-k group compares to that of the top-k group.

**Questions:**

Please see the Weaknesses section for major questions. Here, I detail miscellaneous questions or suggestions that may not be relevant to a weakness.

1. On page 1, `` should be used to write a left quotation mark in LaTeX.
2. The phrase “ordering of the mini-batches”, used on page 1 and Section 4 is confusing. In particular, it is not clear whether each mini-batch contains different data for each model or if the mini-batches contain the same data and only their order is changed.
3. Top-k/bottom-k should be defined in the caption to Figure 1 as the terms have not been introduced in the main paper yet.
4. It would be helpful to denote the raw accuracy percentages in Figures 1, 2, and 3 in addition to the relative increases.

***Recommendation***

While the main observation of this work is potentially interesting and the experiments are comprehensive, the (a) lack of specification in the training and ensembling procedure, (b) lack of contextualization and comparison to the literature, and (c) need for further investigation based on relevant literature motivate my recommendation of a rejection for this submission.

***References***

[1] Sagawa et al. Distributionally Robust Neural Networks for Group Shifts: On the Importance of Regularization for Worst-Case Generalization. ICLR, 2020.

[2] Idrissi et al. Simple data balancing achieves competitive worst-group-accuracy. CLeaR, 2022.

[3] Qiu et al. Simple and Fast Group Robustness by Automatic Feature Reweighting. ICML, 2023.

[4] LaBonte et al. Towards Last-layer Retraining for Group Robustness with Fewer Annotations. NeurIPS, 2023.

[5] Lee et al. Diversify and Disambiguate: Learning From Underspecified Data. ICLR, 2023.

[6] Fort et al. Deep Ensembles: A Loss Landscape Perspective. ArXiv, 2019.

[7] Entezari et al. The Role of Permutation Invariance in Linear Mode Connectivity of Neural Networks. ICLR, 2022.

[8] Jiang et al. Assessing Generalization of SGD via Disagreement. ICLR, 2022.

[9] Lakshminarayanan et al. Simple and scalable predictive uncertainty estimation using deep ensembles. NeurIPS, 2017.

---

### Official Review · Reviewer_9BxN · 2023-10-31

**Soundness:** 2 fair
**Presentation:** 2 fair
**Contribution:** 2 fair
**Rating:** 3
**Confidence:** 3

**Summary:**

Ensembling Deep Neural Networks (DNNs) is simple, yet famous for its high performance. This is because it leverages the collective intelligence of multiple independently trained models to compensate for the mistakes of a single model and increase generalization performance. While many studies have focused on improving top-line metric performance or efficiency, there has been limited research on how sensitive the performance is to specific subsets. From a fairness perspective, the performance of subgroups is commonly considered, and the objective is to mitigate the error rate of a particular subgroup when it is significantly higher than the performance of other subgroups. Other studies have tried to improve fairness by proposing new ensemble methods for this purpose. However, the authors focus on the fairness benefits of the simplest homogeneous ensemble. Surprisingly, they found that a homogeneous ensemble with the same architecture and hyperparameters, thus lacking "diversity," significantly improves the performance of subgroups that would otherwise perform poorly under a single model with sufficiently diverse predictions. Furthermore, they observed that initialization, batch ordering, and data augmentation, not architecture and hyperparameters, can be used to train models that are sufficiently different from each other. The authors verified and analyzed their findings through various experiments.

**Strengths:**

- For fairness, the authors experimentally observed that the simplest homogeneous ensemble significantly improves not only the average performance but also the performance of minority groups without applying other complex methods.
- The authors conducted extensive experiments on various image datasets and model architectures.
- The authors experimentally verified how random initialization, data augmentation realization, and data ordering, which are random sources encountered in deep learning training, affect ensemble performance.

**Weaknesses:**

- I am not an expert on fairness, but the definition of fairness in the paper is a gross oversimplification. As noted in Du et al. (2020) and Caton et al. (2020), it is very difficult to define what is "fair" for an algorithm, but I don't think it's reasonable to say that a homogeneous ensemble improves fairness by simply improving the performance of bottom-k subgroups.
- There is no comparison with other fairness-related methods. Ensembles are generally not easy to use in practical applications unless there is a clear performance advantage, as their cost scales linearly with the number of members. Therefore, it is difficult to see why a deep ensemble should be used over other methods when the improvement in fairness compared to the cost of using a homogeneous ensemble is expected to be large.
- The authors used relative accuracy gain throughout the paper to show the fairness improvement, but I think this has the problem that the numbers can be easily overestimated. For example, if two groups had the same 10%p of performance improvement with ensemble, and their base accuracies were 20% and 80% respectively, the relative accuracy gains would be 50% and 12.5% respectively.
- In addition to the above issue, ensemble does not reduce the performance difference between subgroups much. According to the ResNet50 / CIFAR100 results in Table 1, by ensembling 20 models, the accuracy difference between the top-k and bottom-k groups was reduced from 41.7%p to 35.3%p, which I do not think is a significant improvement.
---
**Du et al.** [Fairness in Deep Learning: A Computational Perspective](https://arxiv.org/abs/1908.08843). *IEEE Intelligent Systems,* 2020

**Caton et al.** [Fairness in Machine Learning: A Survey](https://arxiv.org/abs/2010.04053). *ACM Computing Surveys*, 2020

**Questions:**

- It would be nice to see a comparison of how homogeneous ensemble behaves compared to heterogeneous ensemble (with different architecture and hyperparameters).
- Can the observation in Sections 4.2 and 4.3 be generalized? It seems that all three plots have different trends based on Figure 6.